# Investigation on Vibration Characteristics of Thin-Walled Steel Structures under Shock Waves

**DOI:** 10.3390/ma16134748

**Published:** 2023-06-30

**Authors:** Zehao Li, Wenlong Xu, Cheng Wang, Xin Liu, Yuanxiang Sun

**Affiliations:** 1Institute of Advanced Technology, Shandong University, Jinan 250061, China; 2State Key Laboratory of Explosion Science and Technology, Beijing Institute of Technology, Beijing 100081, China

**Keywords:** thin-walled structures, shock waves, shock tube, structural shock dynamic

## Abstract

Thin-walled steel structures, prized for their lightweight properties, material efficiency, and excellent mechanical characteristics, find wide-ranging applications in ships, aircraft, and vehicles. Given their typical role in various types of equipment, it is crucial to investigate the response of thin-walled structures to shock waves for the design and development of innovative equipment. In this study, a shock tube was employed to generate shock waves, and a rectangular steel plate with dimensions of 2400.0 mm × 1200.0 mm × 4.0 mm (length × width × thickness) was designed for conducting research on transient shock vibration. The steel plate was mounted on an adjustable bracket capable of moving vertically. Accelerometers were installed on the transverse and longitudinal symmetric axes of the steel plate. Transient shock loading was achieved at nine discrete positions on a steel plate by adjusting the horizontal position of the shock tube and the vertical position of the adjustable bracket. For each test, vibration data of eight different test positions were obtained. The wavelet transform (WT) and the improved ensemble empirical mode decomposition (EEMD) methods were introduced to perform a time-frequency analysis on the vibration of the steel plate. The results indicated that the EEMD method effectively alleviated the modal aliasing in the vibration response decomposition of thin-walled structures, as well as the incompletely continuous frequency domain issue in WT. Moreover, the duration of vibration at different frequencies and the variation of amplitude size with time under various shock conditions were determined for thin-walled structures. These findings offer valuable insights for the design and development of vehicles with enhanced resistance to shock wave loading.

## 1. Introduction

Thin-walled steel structures are commonly utilized in various applications such as ships, aircraft, and vehicles due to their lightweight nature, material efficiency, and favorable mechanical properties [1,2]. These structures are often subjected to transient shock loads during operation in challenging environments. As a result, the dynamic response characteristics of thin-walled structures under transient shock loads have received extensive attention from researchers and engineers in related fields [3,4].

The investigation of the dynamic response of thin-walled structures has predominantly focused on two distinct modes of failure. Specifically, the response of these structures to shock loads is generally characterized by either large plastic deformation [5,6,7,8] or tensile tearing [9,10]. For the study of large plastic deformation of thin-walled structures, Wang et al. [11] conducted experimental and numerical simulation studies on the response of free thin steel plates under intense loads and proposed a theoretical model to describe the deformation of free metal thin plates under powerful loading. They found that when the loading parameters are determined, the deformation velocity and deflection of the plate are only related to the width-to-thickness ratio of the plate. Xu et al. [12] presents experimental and numerical investigations on the response of thin aluminum plates to shock loading, reveals the relationship between the deformation region of counterintuitive behavior (CIB) and loading and geometric parameters of the structure, and derives a relationship between normalized duration and charge mass to predict the occurrence of CIB. Kaufmann et al. [13] introduced the Virtual Fields Method (VFM) to reconstruct surface pressures on thin steel plates by measuring full-field deformation of plate dynamics. A shock wave similar to an explosion is generated by a shock tube, demonstrating the crucial role of VFM modeling in predicting large plastic deformations of structures. Curry et al. [14] investigated the impact of different charge backing types on the plastic deformation of thin-walled steel plates using a combination of experimental and computational methods. The findings suggest that metal-backed charges increase impulse transfer by 3–5 times compared to air-backed charges. While the permanent deflection was greater with metal backing, the degree of increase was less pronounced compared to that of the impulse transfer. Kim et al. [15] studied the influence of the relative position of explosives and thin steel plates on the plate damage and found that tearing damage can be reduced by optimizing the inclination angle of the thin wall. Yao et al. [16] conducted a large-scale experimental study on the damage characteristics and dynamic response of thin-walled multi-steel box structures under restrained high-intensity loads and found that strengthening the corner of the box can prevent tearing of the thin-walled structure. McDonald et al. [17] examined the response of four high-strength thin-walled structures to localized blast loading. The findings indicate that higher strength steels and tailored microstructures provide enhanced rupture resistance. A new non-dimensional impulse correction parameter is introduced for assessing the impact of charge stand-off on deformation and rupture performance.

As the level of intelligence and informationization of vehicles increases, electronic and mechanical equipment becomes increasingly important in the carriers [18]. In the face of enemy attacks, even if the structure remains intact, strong vibrations can damage electronic and mechanical devices, thus affecting weapon and equipment effectiveness [19]. Research on the vibration characteristics of thin-walled structures mainly focuses on numerical simulations [20,21,22] and theoretical analysis [23,24,25].

In terms of numerical simulations, Park et al. [26] conduct a numerical investigation to evaluate the effectiveness of thin-walled panels in attenuating vibrational damage induced by explosive events. Their findings reveal that the implementation of blast-resistant panels significantly mitigates the propagation of acceleration, with the most favorable outcomes achieved through the utilization of thicker panels and lower explosion loads. In a separate study, Wu et al. [27] inspect the vibrational response of subterranean thin-walled structures subjected to surface blast loads by employing numerical simulations. The researchers observe an increase in peak velocity as the proximity to the explosion source decreases, with a predominance of vertical vibrations. Moreover, the study introduces a predictive model for damage assessment and delineates critical thresholds associated with distinct damage levels. Wu et al. [28] use numerical simulations to propose a data-driven approach for designing distributed Dynamic Vibration Absorbers (DVAs) to mitigate vibrations in thin-walled structures with tight modal spacing. Leveraging Singular Value Decomposition (SVD) on structural response data, without needing excitation or structural mode info, optimal DVA placement and parameters are determined. This method surpasses traditional techniques, showcasing its robustness and effectiveness on a simply supported square plate and a fairing, achieving broad-band vibration suppression utilizing only structural response data.

For theoretical studies, Pandey et al. [29] investigates the transient vibroacoustic response of functionally graded sandwich plates with varying thickness ratios and material gradation. A parametric study is conducted to investigate the influence of volume fraction index and thickness ratio on the transient vibroacoustic response. Vieira et al. [30] introduced a new high-order beam model for thin-walled structure response analysis, which considers the three-dimensional displacement characteristics of the thin-walled structure and the in-plane bending characteristics of the section. This model effectively analyzes local and global buckling phenomena of thin-walled structures under high-order modes. Xu et al. [31] found that the improved beam theory based on the Carrera Unified Formulation has higher reliability and accuracy in predicting the modal behavior of the structure than the classical beam theory. The complex interactions between shock waves and structures lead to substantial analytical challenges, making it difficult for theoretical analysis to effectively address such complexities [32,33]. With these factors in mind, the importance of experimental research must be highlighted, as it offers valuable insights and enables accurate validation and refinement of theoretical results [34,35].

In this work, an experimental device comprising a shock tube system and a steel plate fixed onto an adjustable support bracket is proposed. The device enables the application of transient shock loads at various positions along a thin-walled structure, facilitating the measurement of its vibration response under shock loading conditions. By conducting transient loading at nine different positions on the structure and using accelerometers at key positions, the transient shock vibration response characteristics were obtained. To address the frequency domain discontinuity problem inherent to traditional wavelet transformation methods, this study proposes a novel empirical mode decomposition method grounded in piecewise cubic Hermite interpolation (PCHIP), an approach that ensures monotonicity and circumvents the “over envelope” and “under envelope” fitting issues commonly associated with conventional interpolation methods, thereby enhancing the analysis of the vibration response of thin-walled structures under varying shock positions.

## 2. Experimental Design and Conditions

### 2.1. Experimental Setups

In this study, transient loads were generated using a shock tube system, as illustrated in Figure 1a. The shock tube, with an inner diameter of 90.0 mm, was partitioned into high-pressure and low-pressure sections, separated by a 0.5 mm-thick aluminum diaphragm with a 0.3 mm cross-shaped scratch. High-pressure nitrogen served as the driving gas in the high-pressure section, while the low-pressure section was initially connected to the atmosphere. When the pressure of the high-pressure section reaches about 1300.0 kPa and the low-pressure section maintains atmospheric pressure, the diaphragm ruptured rapidly along the scratch, creating a shock wave in the low-pressure section. A pressure sensor (Kistler 211B4) was installed at the shock tube opening to measure the pressure generated by the shock wave from the tube. In order to investigate the shock response of thin-walled structures, an adjustable bracket was designed and installed. The bracket featured an adjustable track that allowed for easy modification of the shock wave position by adjusting the plate’s vertical and horizontal positions. A steel plate (measuring 2400.0 mm × 1200.0 mm × 4.0 mm in length, width, and thickness) was mounted onto the bracket. This steel plate is made from Q235 steel, with a density of 7.85 g/cm^3^, a minimum yield strength of 235 MPa, and a Young’s modulus of 210 GPa.

The plate’s response to shock wave loading was analyzed using eight Kistler 8776B100A accelerometers, which were symmetrically positioned along the horizontal and vertical axes of the steel plate. Acceleration data was obtained by processing the voltage signals captured by these sensors with a Kistler TraNET 408DP data acquisition device. Figure 1b shows the arrangement of the accelerometers, with six (RM1-RM6) placed on the plate’s horizontal axis and two (CM1, CM2) on the thin plate’s horizontal axes. The first transverse sensor was placed 1200.0 mm to the left of the steel plate’s center point, with subsequent sensors positioned every 200.0 mm to the right. Sensors CM1 and CM2, located on the vertical axis, were situated 200.0 mm and 500.0 mm above the center point, respectively. This setup was used to determine the structure’s vibration characteristics. All nine shock positions were located on the right side of the thin plate. Shock positions M1, M2, and M3 were positioned on the horizontal axis of symmetry of the thin plate, lying 200.0 mm, 500.0 mm, and 800.0 mm to the right of the plate’s center, respectively. Above these three shock loading points, shock positions H1, H2, and H3 were found at a height of 200.0 mm on the thin plate’s horizontal axis of symmetry. Below these points, loading positions L1, L2, and L3 were situated at a height of 100.0 mm.

### 2.2. Repetition Verification

To ensure the reproducibility of shock wave loading, a repeatability verification experiment was conducted using a shock tube apparatus. Figure 2 depicts the pressure–time history curves obtained by a pressure sensor at the shock tube nozzle in two separate experiments. The initial peak pressures of the shock waves in the two tests were 129 kPa and 128 kPa, respectively, exhibiting a deviation of 0.7%. These results indicate that the shock wave transient load system used in the experiments demonstrates high consistency.

## 3. Experimental Data Analysis Methods

### 3.1. Wavelet Transform Analysis Method

Wavelet Transform (WT) can show the detailed information of signals in time-frequency domain. The wavelet transform $\left(W_{\psi }f\right)\left(a,b\right)$ of a signal *f*(*t*) is defined as:(1)$$\left(W_{\psi }f\right)\left(a,b\right)=\langle f,\psi_{a,b}\rangle ={\left|a\right|}^{-1/2}\int_{-\infty }^{+\infty }f\left(t\right)\psi^{*}\left(\frac{t-b}{a}\right)\;\mathrm{dt}$$
where *a*, *b ∈ R*, and *α ≠ 0* is called the scaling factor, and *b* is called the translation factor. $\psi \left(\frac{t-b}{a}\right)$ represents a series of basis wavelets determined by *a* and *b*, and $\psi^{*}\left(\frac{t-b}{a}\right)$ represents the complex conjugate of the basis wavelet. The role of WT is to transform one-dimensional impulse response signal data into a two-dimensional matrix, where each row represents the wavelet coefficients at different decomposition scales, and each column represents the different time of the impulse response signal data. Since the WT is discontinuous in the frequency domain, it is not feasible to achieve a fully continuous wavelet transform of signal [36]. To accurately capture the frequency domain characteristics of the structure, we have developed a novel method, namely, the improved ensemble empirical mode decomposition method.

### 3.2. Improved EEMD-HHT Analysis Method

#### 3.2.1. EEMD-HHT Analysis Method

Hilbert–Huang transform (HHT) transform is a time-frequency localization analysis method with strong adaptability, which is suitable for processing and analyzing non-stationary signals. HHT transform consists of EMD decomposition and Hilbert transform, but EMD decomposition has the problem of mode aliasing, that is, the components of different frequency bands in the signal cannot be effectively separated. Wu et al. [37] proposed EEMD decomposition, which can prevent the diffusion of low-frequency modal components by adding white noise and alleviate modal aliasing. After the signal is decomposed by EEMD, the combination of multiple IMF components can be observed, and the analytical signal can be observed by Hilbert transform of the IMF component *c*(*t*) of the original signal. After EEMD decomposition of the signal, a group of IMF components can be obtained. Hilbert transform is performed on each order of IMF components to construct the Analytic signal *z*(*t*):(2)$$z\left(t\right)=c\left(t\right)+jH\left[c\left(t\right)\right]=a\left(t\right)e^{j\Phi \left(t\right)}$$

*H*[*c*(*t*)] is the IMF component after the Hilbert transformation, and *j* is an imaginary unit. The instantaneous amplitude *a*(*t*) corresponding to the analytical signal can be obtained:(3)$$a\left(t\right)=\sqrt{c^{2}\left(t\right)+H^{2}\left[c\left(t\right)\right]}$$

#### 3.2.2. Improved Decomposition Method

In the ensemble empirical mode decomposition (EEMD) method, accurate envelope fitting is essential for signal decomposition. However, traditional cubic spline interpolation methods for fitting extreme points often lead to “over envelope” and “under envelope” issues, which can significantly impact the accuracy of decomposition. In this work, an EEMD method based on piecewise cubic Hermite interpolation (Pchip interpolation) was employed, which effectively resolves the fitting issues encountered in traditional methods. To decompose the signal, we used a low-pass filter to isolate the target frequency band and applied a set empirical mode decomposition based on cubic Hermite interpolation, Figure 3 shows the pchip-EEMD transformation process of real signals.

We then compared the decomposition results obtained using this approach with those obtained from EMD decomposition under identical parameters. As illustrated in Figure 4, the IMF components derived from decomposition show a relatively concentrated center frequency in each order, effectively reducing mode aliasing phenomenon.

To validate the effectiveness of the Pchip-EEMD decomposition, Figure 5 provides a comparison between the improved EEMD decomposition results and the power spectral density estimates of different IMF components obtained via the original EMD decomposition. As illustrated in the figure, the frequency bands associated with each IMF component in the improved decomposition method are less prone to aliasing. Conversely, the EMD decomposition results reveal that most of the frequency bands in the signal are aliased in the IMF1 component. This comparison demonstrates that the Pchip-EEMD method effectively mitigates the mode aliasing phenomenon observed in the original decomposition method.

## 4. Experimental Results and Discussion

### 4.1. Vibration Characteristics of Steel Plates Suffering Higher Region’s Shock

#### 4.1.1. Shock Position of H1

Figure 6 illustrates the wavelet coefficient diagrams of vibration signals captured from different measuring points under the H1 shock condition, which is close to the vertical axis of symmetry of the steel plate. The diagrams of Figure 6a–f correspond to six transverse sensors RM1-RM6, while Figure 6g,h correspond to two longitudinal sensors CM1 and CM2. At the beginning of the loading process, the structure exhibited high-frequency vibration with a relatively high amplitude, which gradually transformed over time. In this context, *f*_L_ represents the frequency with the longest vibration duration, and *T*_L_ denotes the duration of that frequency. Measuring point RM1, which is close to the left boundary, showed no lower frequency vibration below 1000 Hz, and the high-frequency vibration duration was significantly shorter than that of other measuring points. The duration of higher frequency vibration was generally shorter than that of lower-frequency vibration, with an inverse correlation between the frequency and duration. At RM1, *f*_L_ = 2670 Hz, while for RM2-RM6, *f*_L_ was concentrated between 100–250 Hz with a *T*_L_ = 500 ms. For CM1, *f*_L_ = 546 Hz, and the vibration duration below 50 Hz decreased with a decrease in frequency. At CM2, *f*_L_ = 117 Hz, with a *T*_L_ = 470 ms.

#### 4.1.2. Shock Position of H2

Figure 7 depicts the acquired data under H2 shock conditions, The vibration characteristics at each measurement point broadly resemble those under the H1 shock condition. For a given measuring point, as the vibration dissipates, the orange curve in the figure shows that high-frequency vibrations tend to have significantly shorter durations compared to low-frequency vibrations. At measurement point RM1 near the left boundary, *f*_L_ = 2606 Hz, which is the highest frequency with the longest vibration duration among all measurement points within the plate. At measurement point CM1 near the upper boundary of the plate, *T*_L_ = 360 ms and *f*_L_ = 325 Hz. The vibration at measurement points RM2, RM4, RM5, and RM6 within the plate exhibits the same *f*_L_ as under the H1 shock condition, while at measurement points RM3 and CM2, *T*_L_ occurs at *f*_L_ = 99 Hz, which is lower than that of the H1 shock condition (as depicted in Figure 7c,i).

#### 4.1.3. Shock Position of H3

For the H3 shock condition, located to the right of the H1 and H2 conditions, the wavelet coefficient diagram in Figure 8 reveals that the vibration signals at each measuring point have shorter durations across all frequency ranges compared to the H1 and H2 conditions. When the loading is near the edge, the overall amplitude of the plate vibration is smaller, leading to shorter durations of vibration. As shown in Figure 8c, the low-frequency vibration duration and amplitude at measuring point RM3, which is equidistant from the vertical symmetry axis as the loading position, are significantly enhanced compared to the H1 and H2 conditions. At measuring point RM1, *f*_L_ = 1228 Hz, with a *T*_L_ = 64 ms, as shown in Figure 8a. The vibration duration at RM1 is consistently shorter than those of other measuring points across all shock conditions (H1, H2, and H3).

### 4.2. Vibration Characteristics of Steel Plates Suffering Middle Region’s Shock

#### 4.2.1. Shock Position of M1

Under the M1 shock condition, the vibration characteristics of the structure change when the shock is located on the horizontal axis of symmetry of the steel plate. Figure 9 presents the wavelet coefficient diagram of the vibration signals at each measuring point. Due to the loading shock position being close to the center of the structure, the amplitude and duration of the vibration signals at each point are higher, as shown in Figure 9b,c,e,f. The duration of the *f*_L_ at measuring points RM2, RM3, RM5, and RM6 is significantly longer than that of the H1–H3 conditions, where the loading position is above the axis of symmetry. Specifically, at measuring point RM2, the *T*_L_ = 790 ms of the *f*_L_ = 117 Hz vibration signal is 32% higher than the average duration under H1–H3 conditions with the same frequency. Measuring point RM3 exhibits a *f*_L_ = 202 Hz vibration signal with a *T*_L_ = 550 ms, while measuring point RM4 shows a relatively small *f*_L_ = 117 Hz vibration frequency with a *T*_L_ = 305 ms compared to other conditions. At measuring point RM5, the *T*_L_ = 800 ms of the *f*_L_ = 187 Hz vibration signal is significantly longer than that of other shock conditions. Moreover, as shown in Figure 9f, the 100–300 Hz vibration at measuring point RM6 has a higher initial amplitude due to its proximity to the loading position, and the *f*_L_ = 109 Hz vibration signal at this point has a *T*_L_ = 830 ms. As shown in Figure 9h, the vibration signal with the longest duration (*T*_L_ = 310 ms) at measuring point CM1, which is near the upper boundary of the plate, has a frequency of *f*_L_ = 378 Hz and is significantly higher than that of other measuring points under this shock condition.

#### 4.2.2. Shock Position of M2

In the M2 shock condition, for RM1, *f*_L_ = 12,053 Hz, similar to the M1 condition. The vibration at *f*_L_ = 138 Hz in measuring points RM2, RM4, and RM6 has the longest duration and higher amplitude compared to other frequencies under this condition. For RM5, *f*_L_ = 85 Hz, with a *T*_L_ of 660 ms as shown in Figure 10e. At CM1, which is closer to the upper side boundary, *f*_L_ = 566 Hz, as shown in Figure 10g, and this frequency is significantly higher than other measuring points except RM1. At CM2, *T*_L_ = 470 ms and *f*_L_ = 69 Hz, which is lower than the frequency with the longest duration under other conditions.

#### 4.2.3. Shock Position of M3

The vibration duration of frequencies above 250 Hz is prolonged at each measuring point under the M3 shock condition when the shock position is near the right boundary and located on the horizontal symmetry axis of the steel plate. The curve in Figure 11 represents the vibration duration of different frequencies at each measuring point under the M3 shock condition. Compared with the previous 100–250 Hz frequency band, the *f*_L_ at measuring points RM2, RM4, RM5, and RM6 has increased to 500 Hz or above under the M3 shock condition. Under these three shock conditions with the shock position located on the horizontal symmetry axis, the *f*_L_ at measuring point RM1 is consistently above 12,000 Hz. For the M3 shock conditions, the duration of vibrations above 250 Hz in the structure significantly increases, and the amplitude is also enhanced. Taking the third-layer IMF component with a center frequency of around 1000 Hz at the RM6 measurement point as an example, the average instantaneous amplitude of vibrations in the M3 condition is 0.93 g, which is significantly higher than that in the M2 condition at 0.81 g.

### 4.3. Vibration Characteristics of Steel Plates Suffering Lower Region’s Shock

#### 4.3.1. Shock Position of L1

The vibration duration and frequency distribution of the structure under the L1 shock condition, as shown in Figure 12, are similar to those under the H1–H3 and M1, M2 shock conditions. However, one distinct difference is that the vibration energy at each measurement point stays focused within specific frequency ranges for extended durations. Measurement points RM2, RM4, and RM6 exhibit significant vibration at *f*_L_ = 110 Hz and 138 Hz for a *T*_L_ of over 800 ms, while measurement points RM3 and RM5 show concentrated vibration at *f*_L_ = 186 Hz. Moreover, measurement point CM1 displays pronounced vibration at *f*_L_ = 546 Hz.

#### 4.3.2. Shock Position of L2

Figure 13 demonstrates that in the L2 condition, where the shock position is to the right of the vertical symmetry axis, the vibration patterns observed at each measurement point are similar to those in the L1 condition. However, in comparison to the L1 condition, the amplitudes of the vibrations at each measurement point are generally reduced. The vibration energy is concentrated in a few fixed frequencies, and in some measurement points, the *T*_L_ is longer than that observed in the L1 condition. At measurement points RM4 and RM6, the vibrations predominantly occur at *f*_L_ = 138 Hz, with a *T*_L_ of approximately 900 ms. At measurement points RM2, RM3, and CM2, the vibration energy is concentrated at *f*_L_ = 215 Hz, with a *T*_L_ of around 800 ms. At measurement point RM3, which is close to the vertical symmetry axis, the amplitude of the vibrations around 200 Hz abnormally increases, and the *T*_L_ is significantly longer than that observed in the L1 condition. Finally, at measurement point CM1, which is close to the upper boundary, the vibrations are concentrated at *f*_L_ = 638 Hz, with a slightly longer *T*_L_ than that observed in the L1 condition.

#### 4.3.3. Shock Position of L3

In the case of the L3 shock position, which is situated closest to the lower right corner of the steel plate, as depicted in Figure 14, the amplitude of each frequency at various measurement points throughout the structure has exhibited an overall reduction, along with a shortened *T*_L_ compared to the L1 and L2 shock conditions. Specifically, the maximum instantaneous amplitude of the fifth-level intrinsic mode function (IMF) at measurement point RM6 under L2 and L3 shock conditions has been reduced by 28% and 47%, respectively, in comparison to the L1 condition. The frequencies that exhibit the longest *T*_L_ at measurement points RM2, RM3, RM5, and RM6 are all *f*_L_ = 186 Hz, with the duration spanning from 445 ms to 892 ms. For the measurement points RM3 and RM4, which are situated nearer to the vertical symmetric position from the loading position, the amplitude below 200 Hz has been higher and the *T*_L_ has been longer, particularly for measurement point RM3.

### 4.4. Effect of Shock Position on Vibration Characteristics of Steel Plates

Figure 15 depicts the vibration characteristics observed at measuring point RM1. In the figure, the bar chart represents the *f*_L_ at measuring point RM1 under different loading conditions: H-*f*_L_ (above horizontal symmetry axis), M-*f*_L_ (on horizontal symmetry axis), L-*f*_L_ (below horizontal symmetry axis), while the line chart represents the *T*_L_ at measuring point RM1 under different loading conditions. It can be observed that for loading positions closer to the horizontal axis symmetry, the *f*_L_ in measuring point RM1 is above 12,000 Hz in M1–M3 cases. However, for H1–H3 and L1–L3 cases, the *f*_L_ is between 2000–4000 Hz, and for the loading cases closer to the right boundary, the frequency is relatively smaller. Let us define the longest duration frequency as *f*_L_. The *T*_L_ changes of different cases have no obvious regularity, but they are all within 100 ms.

In Figure 16, a bar chart is presented that displays the *f*_L_ of vibration at measuring points RM2- RM6 for various loading cases. Compared to measuring point RM1, which is close to the boundary, these measuring points show a significant reduction in *f*_L_, making them unsuitable for inclusion in the same bar chart. Notably, the results reveal that in the M3 case, except for measuring point RM3, which is symmetric to the loading position, the *f*_L_ detected by all other measuring points are significantly higher. Additionally, in both the H3 and M3 cases, measurement point RM3 exhibits a significantly lower vibration frequency compared to other measurement points at the same horizontal position, given that RM3 is located at an equal distance from the vertical axis of symmetry as the loading position. It is worth noting that for most shock conditions investigated, the frequencies with the longest vibration duration (fL) at measuring points RM3 and RM5 are significantly higher compared to other measurement points.

## 5. Conclusions

This paper presents an experimental device consisting of a shock tube system and an adjustable steel plate, designed to investigate the transient shock vibration response of thin-walled structures. By applying shock loads at various positions and using accelerometers to collect data, a comprehensive understanding of the vibration response characteristics has been achieved. To analyze the vibration signals in the time-frequency domain, the research introduced two methods, the wavelet transform and the EEMD with pchip interpolation. The accuracy of these methods is validated, and the results provide information on the distribution and attenuation of vibration energy across various frequency bands. The study reveals several findings, which can be summarized as follows:

1. Improved Signal Analysis: We have refined time-frequency domain analysis by using the Ensemble Empirical Mode Decomposition (EEMD) method with pchip interpolation. This significantly enhances the accuracy, tackling mode mixing issues in traditional EMD and addressing redundancy and frequency-discontinuity problems in continuous wavelet transform (CWT). These enhancements will aid in accurately characterizing structure vibrations, thereby facilitating the development of more shock-resistant designs.

2. Boundary Vibration Attenuation: Our findings show that the boundary regions exhibit the fastest rate of vibration attenuation. This could guide the design and placement of sensitive components within a structure to minimize sustained vibrations.

3. Frequency Band Implications: The research reveals that the 100–300 Hz frequency band sustains the longest vibration duration for most shock conditions, providing pivotal knowledge for predicting potential failure areas or developing frequency-specific vibration dampening methods.

4. Influence of Shock Position: The study underscores that shock positioning significantly impacts vibration characteristics, and this insight can be incorporated into design strategies for specific shock or load conditions.

5. Asymmetric Shock Impact: We found that as the shock position moved towards the right side, the vibration amplitude and duration decreased. This implies that asymmetric shock loads can influence the overall vibrational characteristics, which is a critical consideration for improving equipment resilience and performance.

## Figures and Tables

**Figure 1 materials-16-04748-f001:**
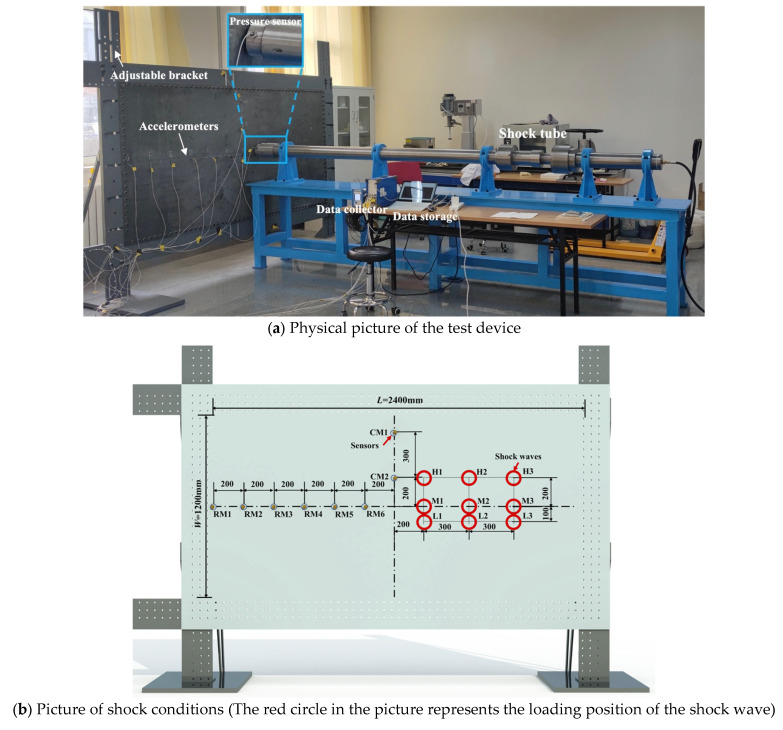
Experimental schematic.

**Figure 2 materials-16-04748-f002:**
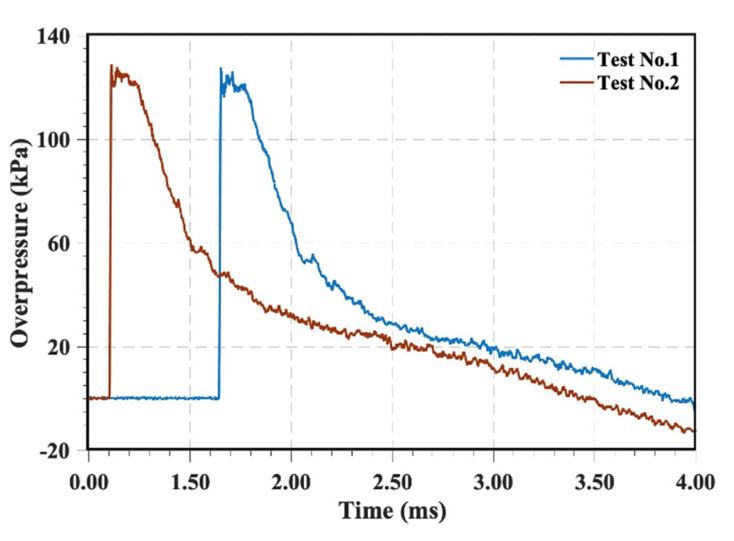
Independent experimental verification.

**Figure 3 materials-16-04748-f003:**
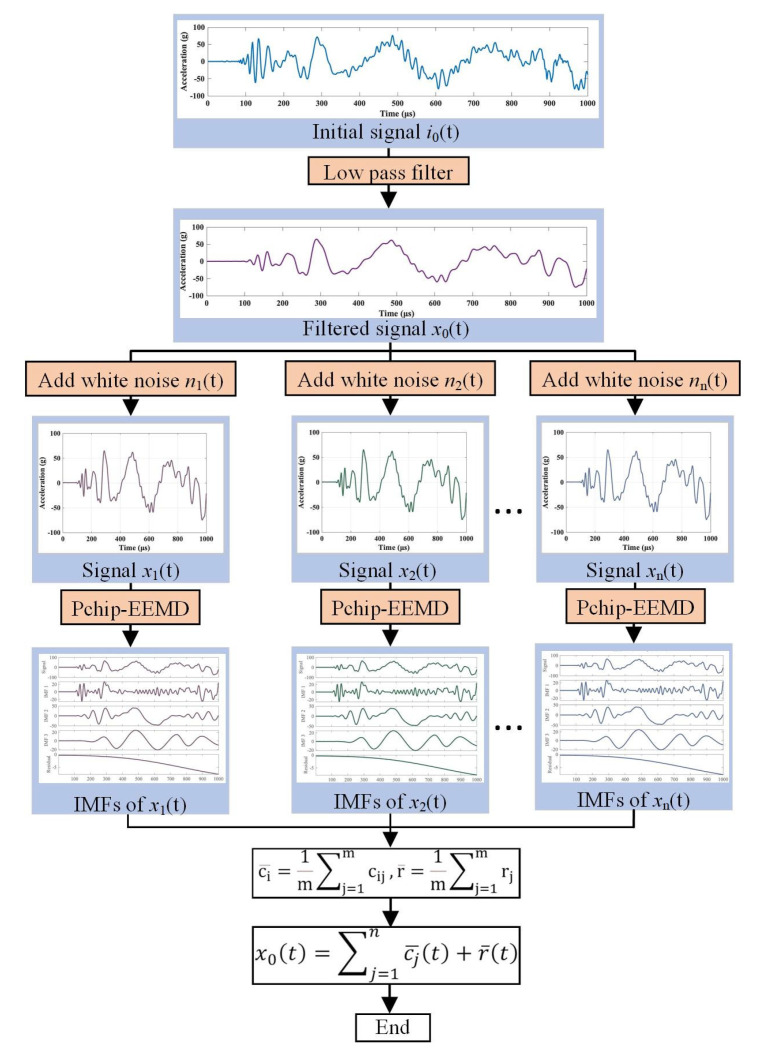
Illustration of the EEMD decomposition principle.

**Figure 4 materials-16-04748-f004:**
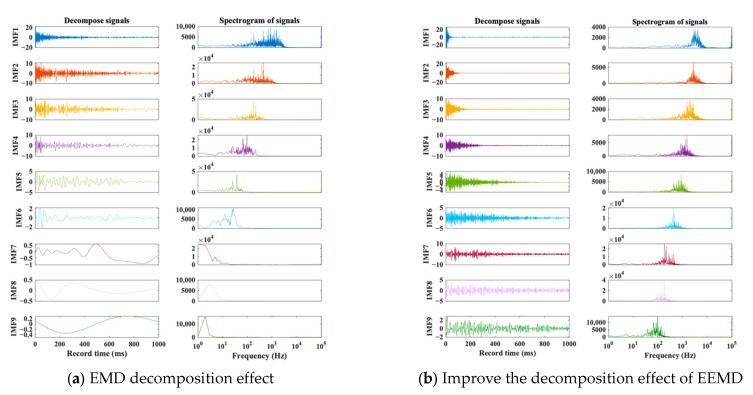
IMF and its spectrum. (Different colored curves represent different IMF components).

**Figure 5 materials-16-04748-f005:**
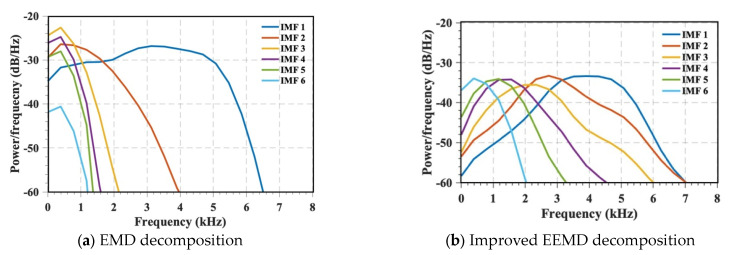
Spectral density estimation of two decomposition methods.

**Figure 6 materials-16-04748-f006:**
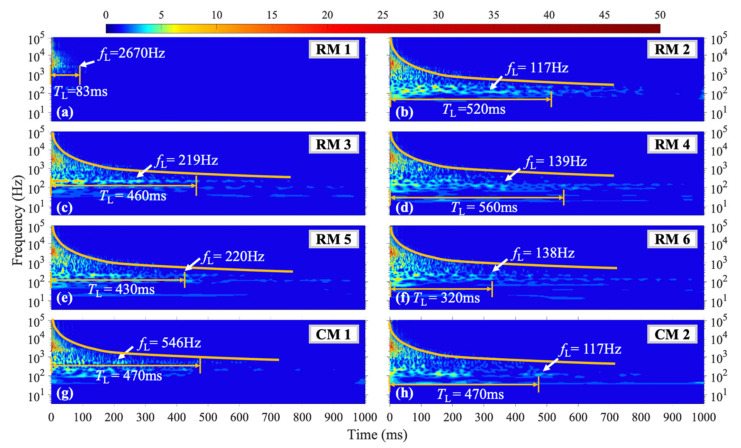
WT coefficient diagram under H1 condition.

**Figure 7 materials-16-04748-f007:**
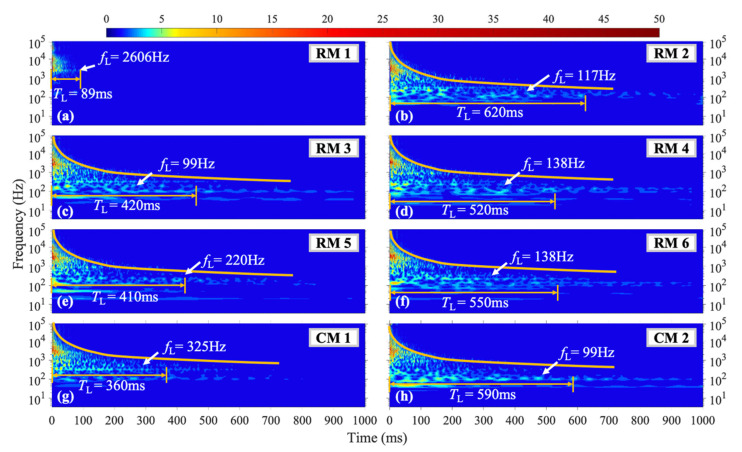
WT coefficient diagram under H2 condition.

**Figure 8 materials-16-04748-f008:**
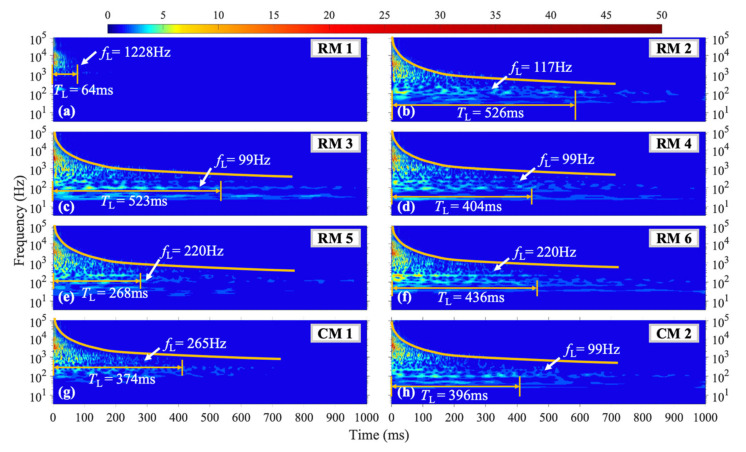
WT coefficient diagram under H3 condition.

**Figure 9 materials-16-04748-f009:**
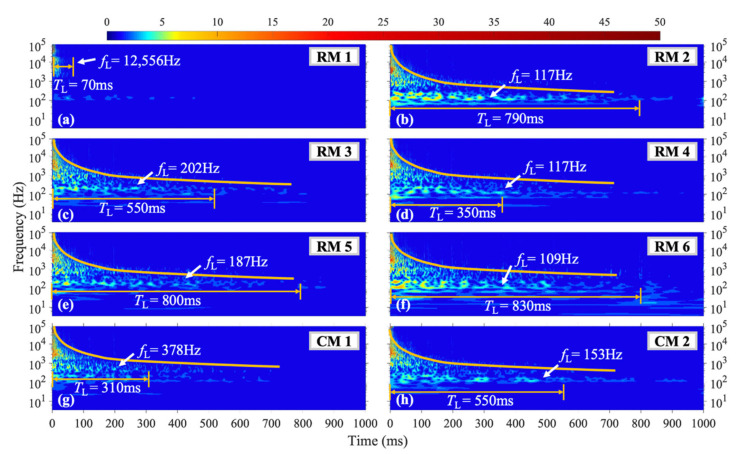
WT coefficient diagram under M1 condition.

**Figure 10 materials-16-04748-f010:**
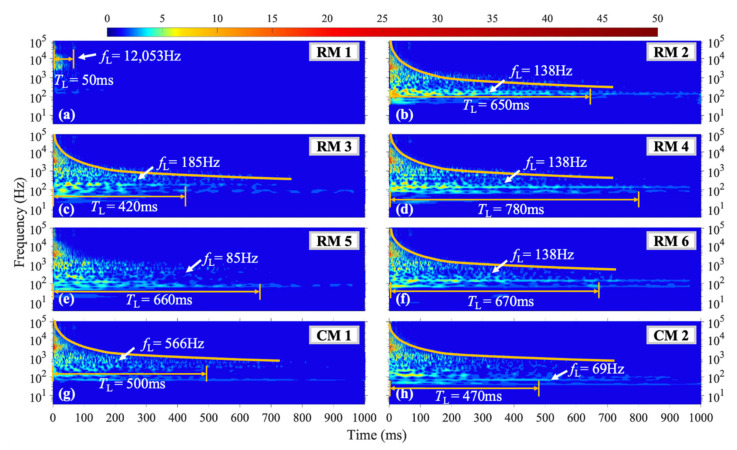
WT coefficient diagram under M2 condition.

**Figure 11 materials-16-04748-f011:**
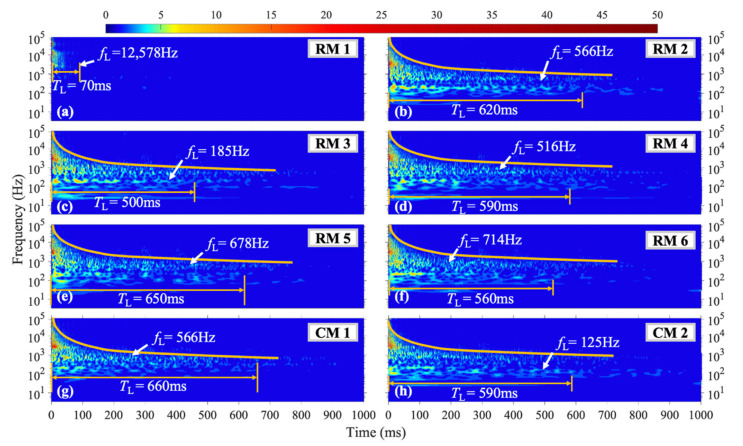
WT coefficient diagram under M3 condition.

**Figure 12 materials-16-04748-f012:**
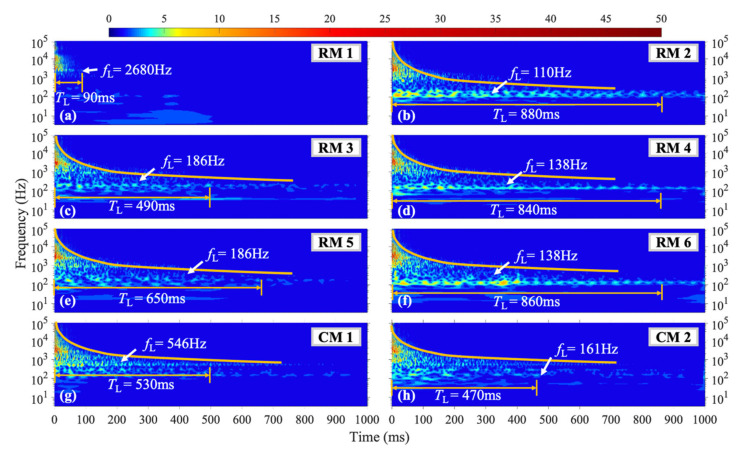
WT coefficient diagram under L1 condition.

**Figure 13 materials-16-04748-f013:**
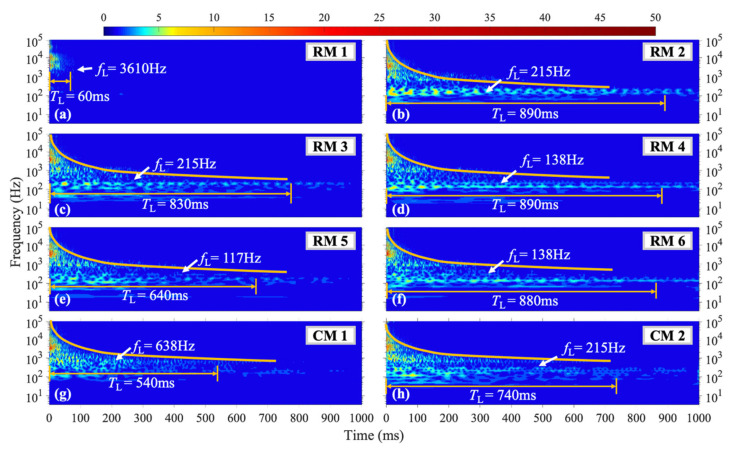
WT coefficient diagram under L2 condition.

**Figure 14 materials-16-04748-f014:**
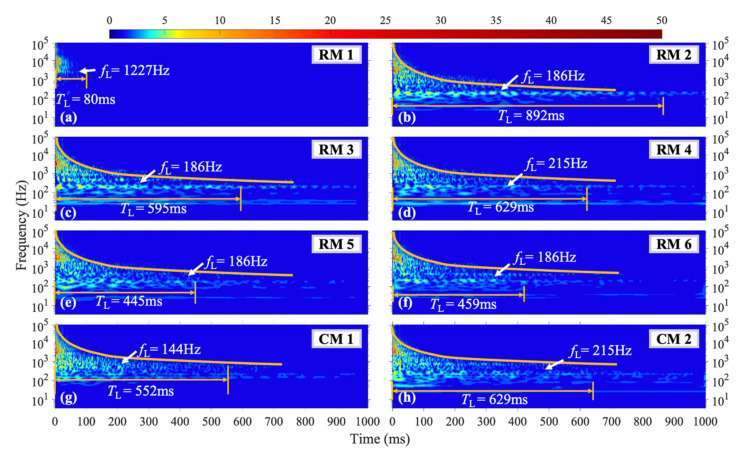
WT coefficient diagram under L3 condition.

**Figure 15 materials-16-04748-f015:**
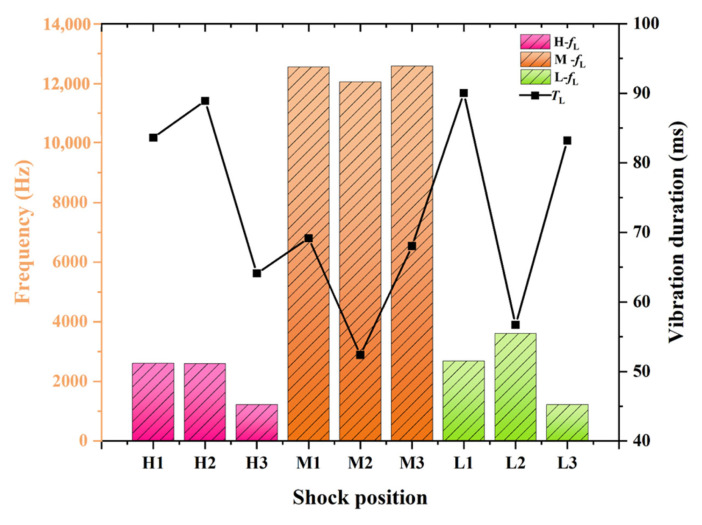
Vibration characteristic of row measuring point RM1.

**Figure 16 materials-16-04748-f016:**
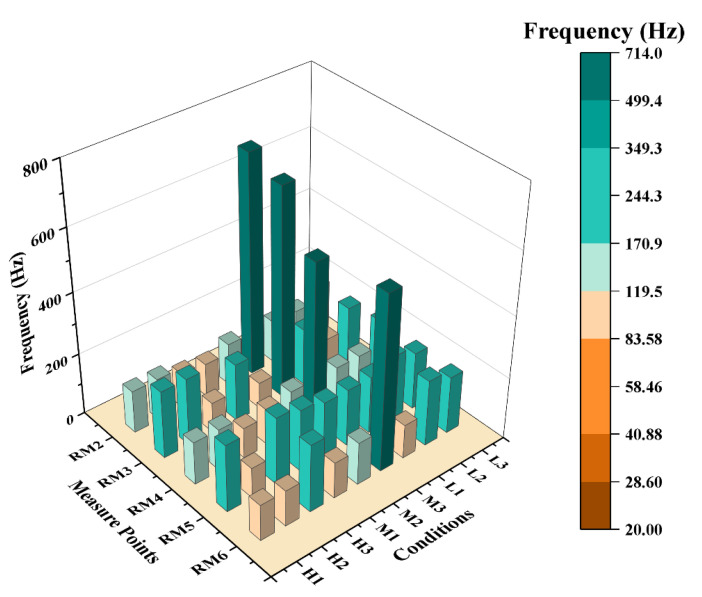
Frequency with the longest vibration duration at each measuring point.

## Data Availability

The data generated or analyzed during this study are included in this published article. For further inquiries, please contact the corresponding author.

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
