# Peer review of "Investigation on Vibration Characteristics of Thin-Walled Steel Structures under Shock Waves"

_materials, 2023, doi:10.3390/ma16134748_

Round 1

Reviewer 1 Report

Dear authors,

I am of the opinion that the paper has good potential. However, it should be improved a bit.

In the paper, you do not specify which steel material it is. It would be good to state what the material is and its mechanical characteristics.

The tests were performed only on one steel plate with a thickness of 4 mm. Why was that particular thickness chosen and what would the results be if a steel plate of a different thickness were chosen?

What is the purpose of this examination and how can this knowledge be applied in practice when determining the dimensions of elements? Is it possible to perform numerical modelling based on this knowledge?

In the conclusion, the results and meaning of these tests should be explained in more detail.

Reviewer 2 Report

The work presented in the manuscript is novel. The following points need to be addressed before publication. 

1) In the introduction states, "Hermite interpolation was proposed". What is the purpose? Write 01 line explanation. 

2) In the introduction, remove the sentence from the chapters. 

3) In the experimental setup, states "was partitioned into high-pressure and low-pressure sections, separated". Provide the approximate pressure values. 

4) Provide the details of steel materials. 

Minor editing of the English language required

Round 2

Reviewer 1 Report

The authors made corrections in the paper according to the instructions of the reviewers and the paper can be published.